# Perceived Barriers of Physical Activity Participation in Individuals with Intellectual Disability—A Systematic Review

**DOI:** 10.3390/healthcare9111521

**Published:** 2021-11-08

**Authors:** Miguel Jacinto, Anabela Sousa Vitorino, Diogo Palmeira, Raul Antunes, Rui Matos, José Pedro Ferreira, Teresa Bento

**Affiliations:** 1Faculty of Sport Sciences and Physical Education, University of Coimbra, 3040-248 Coimbra, Portugal; jpferreira@fcdef.uc.pt; 2Life Quality Research Centre (CIEQV), 2040-413 Rio Maior, Portugal; raul.antunes@ipleiria.pt (R.A.); rui.matos@ipleiria.pt (R.M.); 3Sports Science School of Rio Maior, Polytechnic Institute of Santarém, 2040-413 Rio Maior, Portugal; anabelav@esdrm.ipsantarem.pt (A.S.V.); teresabento@esdrm.ipsantarem.pt (T.B.); 4Research Center in Sport Sciences, Health Sciences and Human Development (CIDESD), 5001-801 Vila Real, Portugal; 5Faculty of Sport, Porto University, 4200-450 Porto, Portugal; diogopalmeira96@gmail.com; 6School of Education and Social Sciences, Polytechnic Institute of Leiria, 2411-901 Leiria, Portugal; 7Center for Innovative Care and Health Technology (ciTechCare), Polytechnic of Leiria, 2410-541 Leiria, Portugal; 8Research Center for Sport and Physical Activity (CIDAF), 3040-248 Coimbra, Portugal

**Keywords:** barriers, intellectual disability, interview, physical activity, sedentary lifestyle

## Abstract

Individuals with intellectual disability (ID) tend to have a sedentary lifestyle, with low physical fitness and an increased risk of chronic diseases. One reason for the prevalence of a sedentary lifestyle is the existence of barriers to participation in physical activity (PA). The purpose of this systematic review is to update knowledge about the perceived barriers of PA participation in individuals with ID. Electronic searches were carried out in the PubMed, Scopus, SPORTDiscus and Web of Science databases, from September 2020 to May 2021, and included articles published between January 2016 and May 2021. The terms used were: “mental retardation”, “intellectual disability”, “intellectual disabilities”, “physical activity”, “motor activity”, “barriers”, “obstacles”, “embarrassment” and “constraint”, in combination with the Boolean operators “AND” or “OR”. After the methodological process, five studies were included for analysis. These studies revealed the existence of several perceived barriers to regular PA participation, which were grouped into five main groups: personal (6 topics), family (4 topics), social (13 topics), financial (1 topic) and environmental (1 topic). The knowledge and identification of participation barriers can be of extreme importance both to institutions and professionals aiming to enhance the participation of individuals with ID in regular PA programs.

## 1. Introduction

Intellectual disability (ID) is characterized by a deficit in intellectual and adaptive functioning in the conceptual, social and practical domain, being identified with deep, severe, moderate and mild degrees, developing before the age of 18 [1].

In this population, sedentary lifestyles prevail [2,3], not meeting the World Health Organization PA guidelines [4].

Due to their sedentary lifestyles, individuals with ID have low levels of physical fitness [5,6,7], with an increased risk of acquiring other comorbidities such as type II diabetes, hypertension, cholesterol and metabolic syndrome [8]. On the other hand, adopting a healthy active lifestyle and regular PA participation positively affect their physical ability (aerobic capacity, strength, balance and flexibility), cognition, health and quality of life [9,10,11,12].

One of the reasons found in the literature that can justify the fact that these individuals adopt sedentary lifestyles is the existence of barriers/obstacles/constraints that make the practice of PA difficult [13,14].

Previous research has already mentioned the existence of these barriers, such as Bossink’s study, which reported that there are 14 personal barriers and 23 environmental barriers to PA participation [13]. Additionally, and according to McGarty and Melville’s study [14], the barriers to PA participation are associated with three main factors: (i) family members, (ii) personal factors and (iii) social factors.

Over the years, the barriers identified by these authors may have already been overcome and new ones may have emerged, and some of these studies are limited in time and fail to analyze other perspectives and perceptions rather than those expressed by family members.

For this reason, the purpose of the present systematic review is to contribute to a better understanding of the perceived barriers of physical activity participation in individuals with intellectual disability, analyze the reasons and the factors involved and to identify the main strategies to be used by professionals based on the perception of the different stakeholders (individuals with ID, their families or technical caregivers).

## 2. Materials and Methods

The systematic review was carried out in accordance with the PRISMA protocol [15,16] and the methods suggested by Bento [17]. The protocol was registered in the INPLASY, with number INPLASY2021100092 (DOI:10.377667inplasy2021.10.0092). The PICOS strategy [18,19] was defined in order to obtain a final sample of studies that: included participants (P) with ID (Down syndrome—DS included), of any age, gender, ethnicity or race; that intended to identify the effects of the perceived barriers to PA (I) on these individuals’ participation in PA (O).

### 2.1. Information Sources and Search Strategy

Exploratory research was performed in the databases (from September 2020 to the 12 May 2021) in order to better understand the potential for this review and to define the research question and methodology to be used. The next day (13 of May 2021), an electronic search was carried out using the following databases: PubMed (all fields), SPORTDiscus, Web of Science and Scopus (article title, abstract and keywords), and this included the period between 2016, i.e., the end date of Bossink et al.’s [13] systematic review, and May 2021. The following search indexed descriptors were used in all databases: “mental retardation” (MeSH Terms), “intellectual disability” (MeSH Terms), “intellectual disabilities” (MeSH Terms), “physical activity” (MeSH Terms), “motor activity” (MeSH Terms), “barriers”, “obstacles”, “embarrassment” (MeSH Terms) and “constraint”, in the following format: (“mental retardation” OR “intellectual disability” OR “intellectual disabilities”) AND (“physical activity” OR “motor activity”) AND (“barriers” OR “obstacles” OR “embarrassment” OR “constraint”). In the first phase, articles were organized and duplicates were identified and excluded using EndNote software. Subsequently, articles were analyzed and selected based on the fulfilment of the defined inclusion and exclusion criteria. In addition, the reference lists were revised and articles of interest were identified and included in the systematic search.

### 2.2. Eligility Criteria

To be included in the present systematic review, studies had to meet the following criteria: (i) full-text scientific publication in the English language; (ii) no restrictions regarding race or ethnicity; (iii) studies with any age group or gender; (iv) studies without restrictions on the number of participants; (v) studies that described the assessment instruments used; (vi) studies that clearly and objectively present the results related to the impact of potential barriers to PA participation in individuals with intellectual disability. The major exclusion criteria used in the study were the following: (i) review articles, comments, theses or abstracts published in minutes of congresses or conferences; (ii) individuals with pathologies other than ID and DS, such as, for example, autism, motor disabilities, hypertension, among others; (iii) studies with athletes registered in sports federations.

### 2.3. Selection and Data Collection Process

After completing the systematic search, duplicates were eliminated and all the articles that did not meet the inclusion criteria were removed. The studies selected in the previous phase were reviewed in their entirety by two independent reviewers (MJ and DP) according to the specific eligibility criteria. The main reviewer (MJ) identified the relevant information about each one of the studies and organized it in summary tables by: authorship, year of publication, country (origin of the research team), objectives, participants, type of study, evaluation techniques, main results/conclusions and quality of information index (see Table 1). After reading the full text of the studies, and according to the eligibility criteria previously defined, the study sample was constituted by five studies.

### 2.4. Evaluation of the Quality of the Studies

The Downs and Black Scale [25] was used to assess the methodological quality of studies. This scale consists of 27 items, punctuated with “one value” or “zero”, characterizing the different parts of an article. The methodological quality of studies was independently assessed by two researchers (MJ and DP). The results obtained by both were compared and discussed, so that a consensus was reached. When consensus was not possible, a third researcher was invited to collaborate (AV). The scale’s scoring intervals received corresponding levels of quality: excellent (26–28); good (20–25); fair (15–19); and poor (≤14). However, as fifteen questions (questions 8, 10–12, 14–17 and 21–27) were not applicable to all studies analyzed, they were removed. The scale, after being modified, had a maximum of 12 points in relation to the original.

In the present study, no study was excluded due to a low quality score.

## 3. Results

### 3.1. Selection of Studies

The initial search carried out in the four databases revealed a total number of 159 articles identified. In the first phase, and after reading the titles and abstracts, seven potentially relevant studies were identified for the next phase. Considering the applicability of inclusion and exclusion criteria previously defined for this systematic review, and after the complete reading of the articles, a sample of five studies was considered for full analysis.

Figure 1 represents the PRISMA flowchart diagram for the selection of studies in this systematic review.

### 3.2. Methodological Quality

The methodological quality of the studies was assessed as poor in all studies; however, no study was excluded due to the low quality score. The quality classification is shown in Table 1.

### 3.3. Characteristics of the Studies

Table 1 presents the characteristics, results and methodological quality of the studies included for final review.

### 3.4. Origin

Through the systematic review process, we identified five studies: three other studies from Europe (Italy [20], United Kingdom [21,22]), one from Oceania (Australia [23]) and another from North America (USA [24]). Although all studies use a qualitative methodology, we can see different designs.

### 3.5. Type of Studies

Two exploratory studies, two qualitative studies and one cross-sectional study were included.

### 3.6. Participants

From a total of 181 participants, only 56 were individuals with ID. All the others participants were family members, technical caregivers or project leaders. Three studies underlined the importance of self-reported responses by individuals with ID [21,23,24]. All studies emphasize the importance of the perception of PA participation barriers in individuals who are supported by institutions that support people with disabilities. At the same time, some studies analyzed facilitators and recommendations for participating in PA. In terms of age group, one of the studies does not refer the age of the individuals with ID [21], three other studies have samples from young adolescents [20,22,24] and one study is focused on elderly participants [23].

### 3.7. Evaluation Techniques

All studies used as instrument or evaluation technique, a semi-structured or rigid interview; however, in two studies we do not have information on the questions or topics. Cartwright et al. [21] used different topics for different groups: seven for individuals with ID themselves, seven for family members and caregivers and four topics for the project leaders. McGarty et al. [22] used four main topics in the semi-structured interview (to family members). Stanish et al. [24] chose to use the questionnaire and passed it on to the participants in the form of a rigid interview, which is easy to answer, in both groups of participants. Two studies did not report the duration of the interview, which lasted a maximum of 55 min in two studies [20,22] and 20 min in another study [24].

### 3.8. Barriers to PA Pratice

Taking into account the results of the studies of this systematic review, we can classify the following barriers to PA participation into five different factors, as shown in Table 2.

Complementarily to previous results, in a study assessing ID athletes’ perceptions about barriers to PA participation [26], social barriers were also identified, namely the lack of adapted transport, in addition to environmental barriers specific to their condition such as travel time to the training site and the time it takes to arrive at the next appointments. As athletes, they were expected to adopt more active lifestyles and have already overcome a set of a posteriori barriers.

Regarding the perception of PA participation barriers in individuals with ID that received support from institutions, they can be divided by personal, social and environmental factors (Table 3), which we highlight:

On the other hand, Table 4 shows the PA participation barriers from the perspective of the interviewed family members.

Analyzing the technical caregivers’ perceived PA participation barriers, we highlight, in Table 5, the following barriers:

The present study also included an article that analyzed the perception of project leaders about PA participation barriers in the population with ID [21], stating that such barriers are related to the following social factors, as shown in Table 6.

The studies included in this systematic review also provide some recommendations for reducing and attenuating barriers, which are presented in Table 7.

## 4. Discussion

The major purpose of this study is to increase knowledge for a better understanding of the perceived barriers of physical activity participation in individuals with ID, identify and discuss the reasons and factors associated with those participation barriers and to identify appropriate strategies to be used by professionals based on the perception of individuals with ID, their families and their technical caregivers.

The social barriers to PA participation are those that present a greater set of topics. In the studies included in the present systematic review, individuals with ID themselves and project leaders are the groups that perceive the least barriers, unlike family members and technical caregivers. In the same sense, a behavioral change in the direct support from professionals is suggested in order to promote PA among individuals with ID [27], increasing interpersonal interaction between both stakeholders as well as the commitment to encourage, adopt and maintain PA participation [28]. On the other end, family members recognize that they are the main barrier to PA participation, since they describe themselves as overprotective of their children, given their characteristics [20]. They must mitigate these attitudes and stimulate PA participation since PA patterns in childhood are seen as relevant predictors of PA participation in adulthood [29,30]. They are preponderant in the process of starting and adhering to PA participation in its quantity, duration and complexity [31,32].

We also highlight those topics such as the characteristics of the disability itself, lack of spaces and adapted activities, which are referenced by the various studies included in the systematic review. The previous reported results are transversal to other types of disability. For motor disabilities, Jaarsma et al. [33] highlighted the following barriers to PA practice: (i) characteristics of the disability itself; (ii) health; (iii) lack of facilities for the practice of PA; (iv) transportation; (v) accessibility. Additionally, Marmeleira et al. [34] identified a set of barriers in the visually impaired population: (i) problems on sidewalks; (ii) lack of adapted facilities; (iii) lack of support from public entities; (iv) need for a guide; (v) lack of adapted PA supply; (vi) lack of security in existing facilities. Tsai and Fung [35] reported the following barriers to PA participation in the hearing-impaired population: (i) uncomfortable feelings with society’s negative attitudes towards disability; (ii) lack of adapted information; (iii) physical discomfort; (iv) lack of physical fitness; (v) lack of direction; (vi) interpersonal restrictions; (vii) lack of adapted facilities. These barriers lead us to the conclusion that there is an important need to create physical exercise programs that are as personalized and adapted as possible to individuals’ needs.

Our study included articles from various age groups, including most perceived barriers to PA participation at different ages, with the exception of aspects inherent to aging, climate, sensory issues and limited human resources that are only identified by the elderly population [23].

The results of the present study confirm the existence of clear barriers to PA participation in individuals with ID at all levels. Some of them have been identified in the literature for quite a long time, as is the case for Messent et al. [36], where intrinsic and extrinsic barriers are mentioned, or in more recent studies, as is the case for Bossink et al. [13], where 14 personal barriers and 23 environmental barriers were identified, and McGarty and Melville’s [14] study that, despite asking only family members, suggested that barriers to PA participation were related to three factors: (i) family members; (ii) personal; and (iii) social. All barriers found through the present study have already been identified and mentioned by previous systematic reviews. However, based on Bossink et al.’s study [13], the barriers seem to have attenuated/decreased, since in this systematic review there were no topics such as: (i) individual fears; (ii) lack of motivation; (iii) anxiety on the part of the technicians (fear of doing something wrong). Aiming to increase the regular practice of PA, the results of this systematic review can be seen to indicate that recommendations/strategies are being put into practice, and this may be at the origin of the decrease in some barriers. Some studies analyzed in this systematic review provide a set of strategies/recommendations [20,21,22,24] that can be seen as contributions to mitigate/decrease barriers to PA practice that must be taken into account. This fact may lead individuals to adopt more active lifestyles, which lead us back to the Ecological Model for Health Promotion [37], in a perspective that can support the process of behavioral change and promote health (Table A1). The Ecological Model for Health Promotion [37] emphasizes the importance of social environments for health promotion and requires more active participation by various stakeholders, where the Personal, Interpersonal, Organizational, Community and Public Policy factors have a fundamental role in the structuring, promotion and implementation of PA programs, reducing and attenuating the barriers to the practice of PA and highlighting the very interventional role of the organization in this process.

A limitation of this systematic review is the fact that the studies selected for analysis did not separate the barriers by degree of ID (mild, moderate, severe or profound), because their impact may be differentiated since they require different support and physical inactivity is greater as the degree of ID increases. However, self-reports are important, although we are aware that not all individuals have the capacity to respond. That said, in future studies, it is important to analyze the barriers to the practice of PA at different degrees/levels, even if through the perception of third parties, so that the support/strategies/interventions are the most adapted possible.

Accordingly, the analysis of barriers to the practice of PA, by age and gender, should also be taken into account in future studies. No studies were found with the Portuguese population, which should be the object of study in future works. Future works should also analyze barriers to the degree of disability (mild, moderate, severe and profound ID) separately.

## 5. Conclusions

The disabled person faces a set of barriers to the practice of PA. Specifically in ID, the main barriers to PA practice, perceived by individuals with ID themselves, their families, caregivers/technicians or even from the perspective of project leaders, can be systematized into personal factors (6 topics), family members (4 topics), social (13 topics), financial (1 topic) and environmental (1 topic).

Since the last published systematic review, the number of perceived barriers to the practice of PA by individuals with ID has decreased. This fact may presuppose basic work carried out, taking into account the strategies and recommendations that have been presented, aiming to promote the practice of PA and therefore influencing this change.

The present work reinforces the existence of a set of barriers to the practice of PA by several interested parties, being a useful tool for researchers and professionals in the process of structuring, promoting and implementing PA programs among individuals with ID, which should be as adapted as possible to the individual and their preferences in order to contribute to an increase in healthy lifestyles and to an improvement in physical fitness, health and quality of life.

## Figures and Tables

**Figure 1 healthcare-09-01521-f001:**
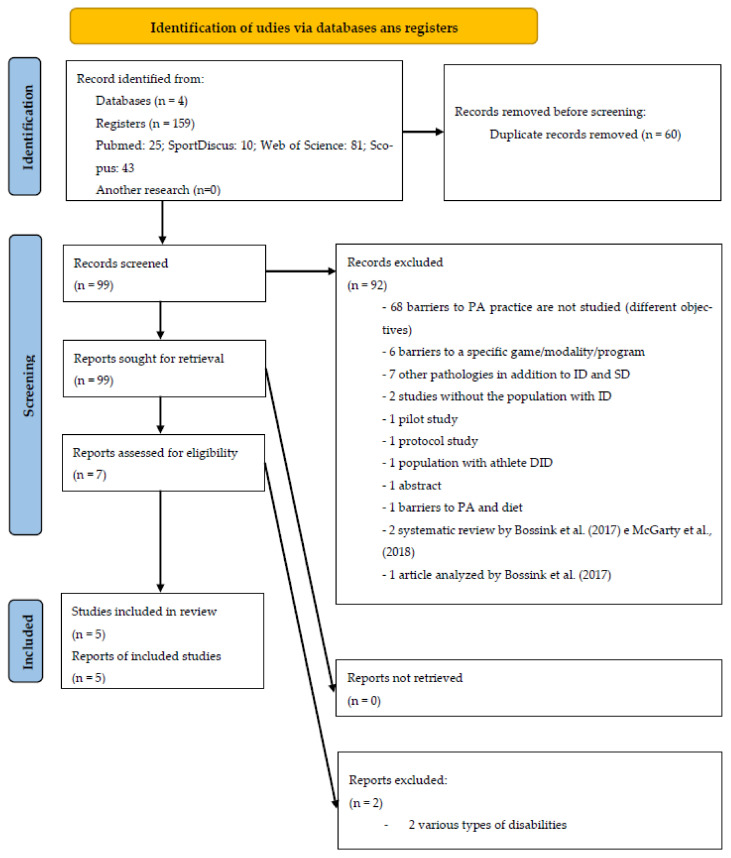
PRISMA flow diagram illustrating each phase of the search and selecting process.

**Table 1 healthcare-09-01521-t001:** Characteristics of the studies.

Author, Reference, Country	Aims	Participants	Type of Study	Evaluation Techniques	Barriers to PA Practice	Quality Score
Alesi [20] Italy	Compare the perceptions about the practice of PA between parents of children with and without DS.	19 families of individuals with DS (children: 10 boys and 9 girls, 20.94Y); Recruitment: support institutions for people with DS.	Exploratory study.	Semi-structured interview with family members; Maximum duration of 25 min.	Lack of technical specialists in adapted PA; Lack of adapted PA programs; Lack of inclusive programs; Characteristics of the disability itself (physical, physiological and psychological); Parent’s preoccupation; Lack of time for parents to engage in PA with their children; Transport difficulties (high costs, lack of transport); Limitation on economic resources.	Poor
Cartwright et al. [21] United Kingdom	Analyze the perspectives of individuals with ID and their caregivers about PA.	N = 42 (12 individuals who were part of the project, 10 family members, 10 technical caregivers and 10 individuals with ID); Recruitment: day centres in Scotland.	Qualitative study.	Semi-structured interviews, with different questions for the 4 groups.	Acceptance of the inactive lifestyle of people with ID—technical caregivers, family members and individuals with ID; Limitation of human resources—technical caregivers and project leaders;Other preferences of intitution-techical caregivers; Communication problems between caregivers and family members regarding the dynamization of PA—technical caregivers, family members and project leaders.	Poor
McGarty et al. [22] United Kingdom	Explore parents’ experience in promoting PA to their children with ID.	N = 8 family members (4 mothers, 3 fathers and a stepfather—a recruited couple, who responded separately); Age of children: 10 to 18 years old (6 male; 1 female) Recruitment: support schools and clubs for people with ID in Glasgow.	Exploratory study.	Semi-structured interviews with family members; Duration: 20 to 55 min.	Lack of information about adequate and inclusive PA; Social exclusion; Fear of parents in relation to bullying; Other preoccupation of parents; Lack of support; Lack of inclusive opportunities; Stigma and lack of understanding about disability; Barriers related to disability itself.	Poor
Salomon et al. [23] Australia	Perception of barriers and facilitators to the practice of PA and healthy eating (separately).	N = 14; 6 renumbered workers and 8 people with ID; ˃60Y; Recruitment: support service for people with ID.	Qualitative study.	Semi-structured interviews.	Both groups: (i) aging; (ii) health problems; (iii) lack of adapted spaces; (iv) lack of inclusion; ID group: (i) chronic diseases; (ii) climatic conditions; Group of workers (i) low concentration; (ii) challenging behaviors; iii) social stigma; (iv) lack of adapted places; (v) sensory issues (example: loud music in spaces; (vi) limitation of financial resources; (vii) limitation of human resources.	Poor
Stanish et al. [24] USA	Compare pleasure with PA, perceived barriers, beliefs and self-efficacy between ID and the general population.	N = 98; ID group (N = 38, 3–21 years, AA:16.8y); general population group (N = 60, 13–18y, AA: 15.3y); Recruitment: agencies, organizations and schools to support individuals with ID.	Cross-sectional study.	Structured interview of 33 closed-response items; Duration: 15 to 20 min.	ID group: (i) PA is difficult to learn; (ii) lack of places to practice PA.	Poor

AA: Average age; ID: Intellectual disability; Min: Minutes; N: Participants; PA: Physical activity; Y: years; DS: Down syndrome.

**Table 2 healthcare-09-01521-t002:** Barriers to PA practice divided by different factors.

Personal	Characteristics of the disability itself (physical, physiological and psychological); acceptance of inactive lifestyles; aging; health problems; lack of concentration; challenging behaviors.
Family members	Parents’ concerns (bullying, among others); acceptance of inactive lifestyles; communication problems with technical caregivers; lack of time to engage in PA with their children.
Social	Acceptance of inactive lifestyles; lack of information on adapted PA; lack of adapted PA programs; lack of inclusive opportunities; lack of technicians specialized in adapted PA; lack of places to practice PA; limitation of human resources; other preferences of the institution providing support services; communication problems between family members and caregivers; social exclusion (stigma and lack of understanding in relation to disability); lack of support; sensory issues (music too loud in training places); difficulties in transportation (high costs, lack of transport).
Financial	Limited financial resources.
Environmental	Climate.

**Table 3 healthcare-09-01521-t003:** Barriers to PA practice in the perception of individuals with ID.

Personal	(i) Preference for inactive lifestyles [21]; (ii) Aging [23]; (iii) Health problems [23].
Social	(i) Lack of adapted spaces [23]; (ii) Lack of inclusion [23]; (iii) Lack of places to practice PA [24]; (iv) Lack of adapted PA [24].
Environmental	(i) Adverse weather conditions [23].

**Table 4 healthcare-09-01521-t004:** Barriers to PA practice in the perception of family members.

Personal	(i) Characteristics of the disability itself [20,22].
Social	(i) Lack of specialists in adapted PA [20]; (ii) Lack of adapted PA programs [20]; (iii) Lack of inclusive programs [20,22]; (iv) Difficulties in transportation [20]; (v) Lack of information about adequate and inclusive PA [22]; (vi) Social exclusion [22]; (vii) Lack of support [22]; (viii) Stigma and lack of understanding of disability [22].
Family members	(i) Parents’ concerns [20,22]; (ii) Lack of time for parents to engage in PA with their children [20]; (iii) Acceptance of children’s inactive lifestyles [21]; (iv) Communication problems between family members and caregivers [21].
Financial	(i) Limitation of economic resources [20].

**Table 5 healthcare-09-01521-t005:** Barriers to PA practice in the perception of technical caregivers.

Personal	(i) Aging [23]; (ii) Health problems [23]; (iii) Low concentration capacity [23]; (iv) Challenging behaviors [23].
Social	(i) Lack of adapted spaces [23]; (ii) Lack of inclusion [23]; (iii) Stigma [23]; (iv) Sensory issues [23]; (v) Lack of human resources in institutions; (vi) Acceptance of inactive lifestyles [21,23]; (vii) Other preferences of technical caregivers and institutions [21]; (viii) Communication problems between family members and caregivers [21].
Financial	(i) Limitation of financial resources [23].

**Table 6 healthcare-09-01521-t006:** Barriers to PA practice in the perception of project leaders.

Social	(i) Limitation of human resources; (ii) Communication problems between family members and caregivers.

**Table 7 healthcare-09-01521-t007:** Summary of study recommendations.

Greater participation by families	Alesi [20]; McGarty et al. [22]; Stanish et al. [25]
Creating more adapted sports offerings	Alesi [20]
Organize environments that aim to stimulate sports participation	Alesi [20]
Organizational change, in the sense of giving greater importance to PA in the lives of people with ID	Cartwright et al. [21]
Local authorities or organizations to increase the offer of adapted PA and finance services	Cartwright et al. [21]; Stanish et al. [25]
Pay more attention to the sporting preferences of individuals	Cartwright et al. [21]
Greater cooperation between all parties in order to promote PA	Cartwright et al. [21]
More and better support and information	McGarty et al. [22]
Personal training	Stanish et al. [24]
PA instruction carried out carefully and with quality	Stanish et al. [24]
Development of group activities	Stanish et al. [24]

## Data Availability

Additional data are available upon request to the author for correspondence.

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
