# Peer review of "Perceived Barriers of Physical Activity Participation in Individuals with Intellectual Disability—A Systematic Review"

_healthcare, 2021, doi:10.3390/healthcare9111521_

Round 1
Reviewer 1 Report
First, the manuscript deals with a very interesting topic. People with intellectual disabilities often lack the resources that would enable them to lead healthy lifestyles. It is necessary, therefore, to identify the main obstacles to confront this situation. However, this manuscript presents certain weaknesses that should be carefully considered prior to publication.
- The study population is confusing. Does the target focus on individuals who are institutionalized or who perform activities in institutions?
- A lack of information in the methodology has been detected. The description of the methodology is ambiguous. Authors mention that they have introduced controlled vocabulary, but they do not discriminate between MeSH and other terms.
- The study population is people with intellectual disability and/or Down syndrome, but the authors do not mention other syndromes associated with intellectual disability.
- Please, revise section 2.4. The numerical sequence is incorrect, the information is repeated and methodology and results are mixed.
- Please review the flow diagram tabulation.
- In table 1. When authors indicate the type of study they included 2 exploratory, 2 qualitative and 1 cross-sectional, but in line 161 they classify them as 4 exploratory studies and 1 cross-sectional. Same terminology must be used.
- Part of the results are included in the discussion.
Author Response
Manuscript ID: healthcare-1438992
Dear Editor-in-Chief of Healthcare MDPI Journal and reviewers,
Response to REVIEWER 1
General Comments: First, the manuscript deals with a very interesting topic. People with intellectual disabilities often lack the resources that would enable them to lead healthy lifestyles. It is necessary, therefore, to identify the main obstacles to confront this situation. However, physical activity this manuscript presents certain weaknesses that should be carefully considered prior to publication.
Response: We would like to thank you for the opportunity to submit a revised draft of our manuscript. The comments of the reviewers were of the utmost importance to help clarify and improve our work. We addressed all the issues raised by the reviewers to whom we wish to thank the time and effort dedicated to providing valuable feedback. We believe that at the present moment the manuscript is suitable for publication. All changes are highlighted in the manuscript. Below we provide a point-by-point response to the reviewers’ comments and concerns.
Comment 1: The study population is confusing. Does the target focus on individuals who are institutionalized or who perform activities in institutions?
Comment 2: A lack of information in the methodology has been detected. The description of the methodology is ambiguous. Authors mention that they have introduced controlled vocabulary, but they do not discriminate between MeSH and other terms.
Response: Thank you very much for the comment. The search strategy used in our study combined and search indexed descriptors defined by of Bossink et al. (2017) with Key Medical Subject Heading (MeSH) terms aiming to update the systematic search results.
Comment 3: The study population is people with intellectual disability and/or Down syndrome, but the authors do not mention other syndromes associated with intellectual disability.
Response: We appreciate the suggestion, which we agree with you. We don’t include other syndromes associated with intellectual disability on our manuscript. As I mentioned earlier, our aims were to complete and update the study by Bossink et al., (2017), which also only investigated in people with intellectual disability and/or Down syndrome.
Comment 4: Please, revise section 2.4. The numerical sequence is incorrect, the information is repeated and methodology and results are mixed
Response: It was a mistake on our part, and we are grateful for warning us. We changed the numerical sequence and eliminated the information that was duplicate.
Comment 5: Please review the flow diagram tabulation.
Response: We included the original flow diagram.
Comment 6: In table 1. When authors indicate the type of study they included 2 exploratory, 2 qualitative and 1 cross-sectional, but in line 161 they classify them as 4 exploratory studies and 1 cross-sectional. Same terminology must be used.
Response: Thank you for alerting us to what happened. In fact, the information shown in Table 1 is correct. We have changed the remaining information so that it is in agreement.
Comment 7: Part of the results are included in the discussion
Response: We agree with your comment. Our main results are presented in table 1 and in the discussion part we intend to compare and debate them.
Additional clarifications
We look forward to hearing from you in due time regarding our submission and to
respond to any further questions and comments you may have.
Sincerely, Miguel Jacinto
Email: migueljacinto1995@gmail.com

Reviewer 2 Report
First, I would like to say that I am very thankful to have the opportunity to read this study. The suggestions given in this document are intended to improve your work. The same feedback document will be given to both editors and authors.
Abstract:
- According to the journal’s instructions: “The abstract should be a total of about 200 words maximum. The abstract should be a single paragraph and should follow the style of structured abstracts, but without headings”.
Introduction section:
- It would be interesting if this section could be developed further. Remember to be as precise as possible so that the reader can understand the scope of your work. At times, it seems like a discussion. Restructure and expand the information a little more to present the objective, and the paper will be more complete.
Methods section:
- The authors stated have used an outdated version of PRISMA (reference 17). I recommend these readings so that you can review the manuscript. The PRISMA diagram is correct, according to the 2020 recommendations.
- http://prisma-statement.org/PRISMAStatement/PRISMAStatement
- https://www.equator-network.org/reporting-guidelines/prisma/
- Why did it take the authors 9 months to perform the review, and don't you think it could have affected the results?
- There are two exclusion criteria sections (2.3 and 2.4) with different information. Please examine this and create a single inclusion-exclusion criteria section.
- Were studies of people with ID and comorbidities included or excluded?
- Was the search register on PROSPERO o similar?
- What about the risk of bias assessment?
- What about the effect measures?
Results and discussion section:
- Please provide figure 1 with higher quality.
- You cannot name table 2 before table 1 in the text, it is confusing.
- The discussion is not really a discussion. In the discussion the authors should highlight the main findings of their study, comparing the variables analyzed with those of other studies, and then indicate limitations and future lines. Finding a table in the discussion is confusing.
- Therefore, I would like to point out that the results and discussion sections should be rethought and rewritten, based on the instructions of the journal, and on what is indicated in the PRISMA 2020 statement.
Author Response
Manuscript ID: healthcare-1438992
Dear Editor-in-Chief of Healthcare MDPI Journal and reviewers,
Response to REVIEWER 2
General Comments: First, I would like to say that I am very thankful to have the opportunity to read this study. The suggestions given in this document are intended to improve your work. The same feedback document will be given to both editors and authors.
Response: We would like to thank you for the opportunity to submit a revised draft of our manuscript. The comments of the reviewers were of the utmost importance to help clarify and improve our work. We addressed all the issues raised by the reviewers to whom we wish to thank the time and effort dedicated to providing valuable feedback. We believe that at the present moment the manuscript is suitable for publication. All changes are highlighted in the manuscript. Below we provide a point-by-point response to the reviewers’ comments and concerns.
Abstract:
According to the journal’s instructions: “The abstract should be a total of about 200 words maximum. The abstract should be a single paragraph and should follow the style of structured abstracts, but without headings”. Response: Thank you for your comments, which we took into account according to the journal’s instructions.
Introduction section:
It would be interesting if this section could be developed further. Remember to be as precise as possible so that the reader can understand the scope of your work. At times, it seems like a discussion. Restructure and expand the information a little more to present the objective, and the paper will be more complete. Response: Your comments/suggestions are very interesting and relevant to improve clarification of our work. We have restructure and expand de introduction so that the work gets more complete.
Methods section:
The authors stated have used an outdated version of PRISMA (reference 17). I recommend these readings so that you can review the manuscript. The PRISMA diagram is correct, according to the 2020 recommendations.
http://prisma-statement.org/PRISMAStatement/PRISMAStatement
https://www.equator-network.org/reporting-guidelines/prisma/
Response: Thank you for the suggestion to read.
Why did it take the authors 9 months to perform the review, and don't you think it could have affected the results? Response: The review does not take 9 months to complete. We express ourselves badly, for which we ask forgiveness. Before proceeding with the systematic review, there was a whole previous work, namely several exploratory researches, with the objective of defining exactly the search databases, terms to be used, the PICOS question, etc. This process took until May 12th. On May 13th, the bibliographic search was carried out, extracting articles that met the eligibility criteria. From that moment until the submission of the manuscript in the journal, it was the time we took with the data analysis and writing of the article).
There are two exclusion criteria sections (2.3 and 2.4) with different information. Please examine this and create a single inclusion-exclusion criteria section. Response: It was a mistake on our part, and we are grateful for warning us. We are grateful for your suggestions.
Were studies of people with ID and comorbidities included or excluded? Response: Studies of people with ID and other disability associated were excluded.
Was the search register on PROSPERO o similar? Response: protocol was registered in the INPLASY, with number INPLASY2021100092 (DOI: 10.37766/inplasy2021.10.0092) and the search was carried out exactly as described.
What about the risk of bias assessment? Response: Given the type of studies that make up the sample, the results found and the objective of the systematic review, an analysis of the risk of bias was not carried out. However, in order to reduce this risk, we carried out an exhaustive search, which we considered to be comprehensive.
What about the effect measures? Response: Our systematic review is only qualitative.The extracted studies and their nature don’t allow us to proceed to a quantitativeanalysis.
Results and discussion section:
Please provide figure 1 with higher quality. Response: We included the original flow diagram.
You cannot name table 2 before table 1 in the text, it is confusing. Response: It was a mistake on our part, and we are grateful for warning us.
The discussion is not really a discussion. In the discussion the authors should highlight the main findings of their study, comparing the variables analyzed with those of other studies, and then indicate limitations and future lines. Finding a table in the discussion is confusing.
Therefore, I would like to point out that the results and discussion sections should be rethought and rewritten, based on the instructions of the journal, and on what is indicated in the PRISMA 2020 statement.
Response: Authors agree and appreciate your comment. They are very interesting and relevant to improve and clarify the results and discussion sections and our manuscript. We will review and rewritten these sections.
Additional clarifications
We look forward to hearing from you in due time regarding our submission and to
respond to any further questions and comments you may have.
Sincerely, Miguel Jacinto
Email: migueljacinto1995@gmail.com
Round 2
Reviewer 1 Report
First, changes introduced have significantly improved the work. However, there are some other suggestions that should be considered:
- line 61: Is the objetive divided in two parts? then authors should include "and to analyse"
- line 66-71: Elegibility criteria: When the authors said "The PICOS strategy was defined", participants should be individuals with ID (not with DS ). Please revise also line 71.
- line 74: "Several exploratory articles researches" is not well understood.
- The item "3.8. Barriers to PA practique" is well synthesized, but its presentation could be improved.
- Table 2 should not be included in the discussion. Consider summarizing to add information in this part of the document.
Author Response
Manuscript ID: healthcare-1438992
Dear Editor-in-Chief of Healthcare MDPI Journal and reviewers, We would like to thank you for the new opportunity to submit a new revised draft of our manuscript. The comments of the reviewers were of the utmost importance to help clarify and improve our work. We addressed all the issues raised by the reviewers to whom we wish to thank the time and effort dedicated to providing valuable feedback.
Response to REVIEWER 1
First, changes introduced have significantly improved the work. However, there are some other suggestions that should be considered:
Comment 1: line 61: Is the objetive divided in two parts? then authors should include "and to analyse"
Response: Thank you for your comments, however, the objective divided in three parts.
Comment 2: line 66-71: Elegibility criteria: When the authors said "The PICOS strategy was defined", participants should be individuals with ID (not with DS ). Please revise also line 71
Response: Dear reviewers, was I mentioned earlier, our objectives were to complete and update the study by Bossink et al., (2017), which investigated people with intellectual disabilities and/or Down syndrome. For this reason, our participants were people with intellectual disability and/or down syndrome.
Comment 3: line 74: "Several exploratory articles researches" is not well understood.
Response: it is research that aims, through methods and criteria, to provide information on a little-known or little studies topic and to guide information on the study's hypotheses.
Comment 4: The item "3.8. Barriers to PA practique" is well synthesized, but its presentation could be improved.
Response: Thanks for your comment which we agree with you. So that the presentation could be improved and to facilitate the interpretation of the data, we created tables.
Comment 5: Table 2 should not be included in the discussion. Consider summarizing to add information in this part of the document.
Response: Dear reviewer, table 2 has been insert in the appendix section and the information summarized in the discussion.
Additional clarifications
We hope that the changes we made to the manuscript have met your suggestions and improved its quality. We look forward to hearing from you in due time regarding our submission and to respond to any further questions and comments you may have.
Sincerely, Miguel Jacinto
Email: miguel.s.jacinto@ipleiria.pt

Reviewer 2 Report
The sentence in line 36 is isolated, it should be integrated into the text.
The introduction section still seems more like a discussion than an introduction. Remember that its purpose should be to introduce the variables, describe the problem they address and finally, state the objective. It can be improved. I recommend reading other introductions to systematic reviews.
Eligibility criteria are the same as inclusion-exclusion criteria. Please clarify this error.
Word the PICO question better.
Remember to explain all acronyms.
The numbering in the methods section is wrong. Check them.
The figure is a screen shot with poor quality. Even the word marks are visible. Please provide a higher quality figure without marks.
The table in the discussion still looks weird. I suggest redacting the results and sending that table to the appendix or supplementary material.
I am also not convinced that table 1 is referred to so many times in the text. It makes it difficult to read. Perhaps the authors could split the information to make it easier to read and have each table go just below its callout in the text.
Author Response
Manuscript ID: healthcare-1438992
Dear Editor-in-Chief of Healthcare MDPI Journal and reviewers, We would like to thank you for the new opportunity to submit a new revised draft of our manuscript. The comments of the reviewers were of the utmost importance to help clarify and improve our work. We addressed all the issues raised by the reviewers to whom we wish to thank the time and effort dedicated to providing valuable feedback.
Response to REVIEWER 2
Comment 1: The sentence in line 36 is isolated, it should be integrated into the text.
Response: Thank you for identifying the error. The same has been fixed.
Comment 2: The introduction section still seems more like a discussion than an introduction. Remember that its purpose should be to introduce the variables, describe the problem they address and finally, state the objective. It can be improved. I recommend reading other introductions to systematic reviews.
Response: Dear reviewer, thank you for your suggestions. We read other introductions to systematic reviews. Our introduction changed, in other to simplify and be more objetive, introduce the variables (participants), describe the problem they address (sedentary lifestyle, commorbility associated and one of the reasons for these sedentary lifestyles) and finally, state the objective.
Comment 3: Eligibility criteria are the same as inclusion-exclusion criteria. Please clarify this error.
Response: Tanks your comments. We agree with you and reformulated da eligibility criteria and inclusion-exclusion criteria sections.
Comment 4: Word the PICO question better.
Response: Thanks for your comment. The PICO question was reformulated.
Comment 6: The numbering in the methods section is wrong. Check them
Response: Thank you for identifying the error. The same has been fixed.
Comment 7: The figure is a screen shot with poor quality. Even the word marks are visible. Please provide a higher quality figure without marks.
Response: We included the original flow diagram.
Comment 8: The table in the discussion still looks weird. I suggest redacting the results and sending that table to the appendix or supplementary material.
Response: We agree with you comment, so we have decided insert the table in the appendix section and the information summarized in the discussion.
Comment 9: I am also not convinced that table 1 is referred to so many times in the text. It makes it difficult to read. Perhaps the authors could split the information to make it easier to read and have each table go just below its callout in the text.
Response: Dear reviewer, as we have inserted new tables in the results section to make them easier to understand and well synthesized (reviewer 1 suggestion), we decided not to divide table 1 into other tables in order not to overload the same section.
Additional clarifications
We hope that the changes we made to the manuscript have met your suggestions and improved its quality. We look forward to hearing from you in due time regarding our submission and to respond to any further questions and comments you may have.
Sincerely, Miguel Jacinto
Email: miguel.s.jacinto@ipleiria.pt
